# Robot-Assisted Augmented Reality (AR)-Guided Surgical Navigation for Periacetabular Osteotomy

**DOI:** 10.3390/s24144754

**Published:** 2024-07-22

**Authors:** Haoyan Ding, Wenyuan Sun, Guoyan Zheng

**Affiliations:** Institute of Medical Robotics, School of Biomedical Engineering, Shanghai Jiao Tong University, Shanghai 200240, China; ding-hy@sjtu.edu.cn (H.D.); wenyuansun1998@sjtu.edu.cn (W.S.)

**Keywords:** augmented reality, robot-assisted, surgical navigation, periacetabular osteotomy, computer-assisted orthopedic surgery

## Abstract

Periacetabular osteotomy (PAO) is an effective approach for the surgical treatment of developmental dysplasia of the hip (DDH). However, due to the complex anatomical structure around the hip joint and the limited field of view (FoV) during the surgery, it is challenging for surgeons to perform a PAO surgery. To solve this challenge, we propose a robot-assisted, augmented reality (AR)-guided surgical navigation system for PAO. The system mainly consists of a robot arm, an optical tracker, and a Microsoft HoloLens 2 headset, which is a state-of-the-art (SOTA) optical see-through (OST) head-mounted display (HMD). For AR guidance, we propose an optical marker-based AR registration method to estimate a transformation from the optical tracker coordinate system (COS) to the virtual space COS such that the virtual models can be superimposed on the corresponding physical counterparts. Furthermore, to guide the osteotomy, the developed system automatically aligns a bone saw with osteotomy planes planned in preoperative images. Then, it provides surgeons with not only virtual constraints to restrict movement of the bone saw but also AR guidance for visual feedback without sight diversion, leading to higher surgical accuracy and improved surgical safety. Comprehensive experiments were conducted to evaluate both the AR registration accuracy and osteotomy accuracy of the developed navigation system. The proposed AR registration method achieved an average mean absolute distance error (mADE) of 1.96 ± 0.43 mm. The robotic system achieved an average center translation error of 0.96 ± 0.23 mm, an average maximum distance of 1.31 ± 0.20 mm, and an average angular deviation of 3.77 ± 0.85°. Experimental results demonstrated both the AR registration accuracy and the osteotomy accuracy of the developed system.

## 1. Introduction

Periacetabular osteotomy (PAO) is an effective approach for the surgical treatment of developmental dysplasia of the hip (DDH) [1,2]. By detaching the acetabulum from the pelvis and then reorienting it, the femoral coverage can be improved, which reduces pressure between the femoral head and the acetabular cartilage, alleviating patients’ pain [1,2,3,4]. However, PAO is technically demanding due to the complex anatomical structure around the hip joint and the limited field of view (FoV) during surgery [5]. In conventional PAO surgeries, surgeons have to mentally fuse preoperative images with the patient anatomy and then perform a freehand osteotomy, resulting in low osteotomy accuracy and limited surgical outcomes.

To solve this challenge, computer-assisted navigation systems for PAO surgery have been reported, providing surgeons with both preoperative assistance and intraoperative guidance [4,6,7,8]. Specifically, computer-assisted surgical planning allows surgeons to analyze the femoral head coverage, to define osteotomy planes, and to determine the optimal reorientation angle based on preoperatively acquired computed tomography (CT) or magnetic resonance (MR) images [6]. Additionally, surgical navigation is used during PAO surgery. By visualizing surgical instruments and osteotomy planes with respect to preoperatively acquired images, surgeons are provided with visual guidance during osteotomy and acetabulum reorientation [4,6,7]. Furthermore, with the rapid development of surgical robots [9,10] and 3D printing [11,12], robot assistance and patient-specific templates have been introduced in PAO surgeries, providing physical guidance during the procedure [13,14,15].

In recent years, augmented reality (AR) technology has been employed in computer-assisted orthopedic surgeries (CAOSs) [16,17,18,19,20,21]. Compared with conventional surgical navigation [4,6,7], AR guidance can reduce surgeons’ sight diversion [22,23]. Specifically, by wearing an optical see-through (OST) head-mounted display (HMD), virtual models of surgical planning, patient anatomy, and instruments are superimposed on the corresponding physical counterparts [12,19]. Thus, surgeons no longer need to switch their focus between the patient anatomy and the computer screen, leading to higher surgical safety. In the literature, AR navigation in CAOS can be roughly divided into two categories: inside–out navigation [16,17] and outside–in navigation [18,19,20,21]. For methods belonging to the former category, fiducial markers (such as ArUco markers [16] and Vuforia Image Target [24]) are rigidly attached to both patient anatomy and surgical instruments. Thus, they can be tracked by cameras on the OST-HMD and then aligned with the corresponding virtual models [16,17]. In contrast, outside–in navigation methods utilize an external tracker (such as an optical tracker or an electromagnetic (EM) tracker) [18,19,20,21], which have larger tracking range and higher tracking accuracy than inside–out navigation [25]. By performing an AR registration, a transformation between the tracker coordinate system (COS) and the virtual space COS is estimated [18,19,21]. Then, virtual models can be aligned with the corresponding physical counterparts tracked by the tracker. However, to the best of the authors’ knowledge, previous AR guidance systems for PAO have not been integrated with robot assistance. Thus, despite AR guidance, surgeons still have to perform a freehand osteotomy without any robot assistance, resulting in low accuracy and safety.

To tackle this issue, in this paper, we propose a robot-assisted, AR-guided surgical navigation system for PAO. The main contributions are summarized as follows:We propose a robot-assisted, AR-guided surgical navigation system for PAO, built on a robot arm and a Microsoft HoloLens 2 headset (Microsoft, Redmond, USA), which is a state-of-the-art (SOTA) OST-HMD. Our system automatically aligns a bone saw with a preoperatively planned osteotomy plane and then provides surgeons with both virtual constraints and AR visual guidance, improving surgical accuracy and safety.We propose an optical marker-based AR registration method. Specifically, we control a robot arm to align an optical marker attached to the robot flange with predefined virtual models, collecting point sets in the optical tracker COS and the virtual space COS, respectively. The transformation is then estimated based on paired-point matching.Comprehensive experiments were conducted to evaluate both AR registration accuracy and osteotomy accuracy of the proposed system. Experimentally, the proposed AR registration method can accurately align virtual models with the corresponding physical counterparts while the navigation system achieved accurate osteotomy on sheep pelvises.

## 2. Related Works

### 2.1. Surgical Navigation in PAO

In recent years, surgical navigation has been employed in PAO to provide surgeons with surgical guidance [4,6,7,8]. Liu et al. [6] introduced a computer-assisted planning and navigation system for PAO, involving preoperative planning, reorientation simulation, intraoperative instrument calibration, and surgical navigation for both osteotomy and acetabular reorientation. Pflugi et al. [4] developed a cost-effective navigation system for PAO, utilizing gyroscopes instead of an optical tracker to measure acetabular orientation. Furthermore, Pflugi et al. [7] proposed a hybrid navigation system, combining gyroscopic visual tracking with Kalman filtering to facilitate accurate acetabular reorientation. However, conventional surgical navigation systems for PAO exhibit several limitations: (1) surgeons need to frequently switch their view between the computer screen of the navigation system and the patient anatomy. This can be inconvenient, and it may increase the potential for inadvertent errors. (2) Despite visual guidance, surgeons still need to perform a freehand osteotomy, which may reduce surgical accuracy.

### 2.2. Robot Assistance in Osteotomy

The past few decades have witnessed the rapid development of robot-assisted osteotomy [13,14,15,26]. Sun et al. [13] proposed an EM navigation-based robot-assisted system for mandibular angle osteotomy, where a robot arm is employed to position a specially designed template for osteotomy guidance. Bhagvath et al. developed an image-guided robotic system for automated robotic spine osteotomy [26]. Tian et al. [14] proposed a virtual fixture-based shared control method for curve cutting in robot-assisted mandibular angle split osteotomy. Shao et al. [15] proposed a robot-assisted navigation system based on optical tracking for mandibular reconstruction surgery, where an osteotomy slot was installed on the robot flange to provide physical guidance. However, to the best of the authors’ knowledge, no robot-assisted system has been reported in the literature for PAO. Additionally, although a few robot-assisted augmented reality surgical navigation systems have been reported for osteotomy [15], surgeons still have to switch their focus between the computer screen and surgical area, which may lower surgical safety. Thus, in this paper, we aim to introduce robot assistance to the PAO procedure.

### 2.3. AR Guidance in CAOS

AR technology has significantly contributed to various orthopedic surgeries [16,17,18,19,21]. Liebmann et al. [16] proposed an AR guidance system for pedicle screw placement. Hoch et al. [17] developed an AR guidance system for PAO. Mendicino et al. [12] used AR as a tool to guide the placement of patient-specific templates, aiming for pelvis resections with higher accuracy and lower time cost. These methods used fiducial markers to achieve real-time tracking based on cameras on the HoloLens. In contrast, Sun et al. [18] developed an external tracker-based AR navigation system for maxillofacial surgeries, where the AR registration was performed by digitizing virtual points using a trackable pointer. Tu et al. [19] proposed an AR registration method based on a registration cube attached to an EM sensor, which was applied to EM tracker-based AR navigation for distal interlocking. Furthermore, to enhance the depth perception, Tu et al. [21] proposed a multi-view interactive AR registration method for the placement of guide wires based on AR assistance. However, to the best of the authors’ knowledge, no AR navigation system has been reported to integrate with surgical robots for PAO. Thus, despite visual guidance, surgeons still have to perform surgeries without any robot assistance. Thus, a robot-assisted AR-guided surgical navigation system is highly desired during the PAO procedure.

## 3. Method

### 3.1. Overview of the Proposed Navigation System

An overview of the proposed surgical navigation system setup is illustrated in Figure 1. As shown in the figure, the proposed system consists of an optical tracker (OP-M620, Guangzhou Aimooe Technology Co., Ltd., Guangzhou, China), a Microsoft HoloLens 2 headset, a robot arm (UR 5e, Universal robots Inc., Odense, Denmark), a medical bone saw (BJ5101, Bojin Medical Inc., Shanghai, China), an optical marker, a dynamic reference base (DRB), and a master computer. Specifically, the medical bone saw and the optical marker are rigidly attached to the flange of the robot arm. The thickness of the saw blade that is used in our study is 0.58 mm. The DRB is rigidly attached to the patient anatomy. The master computer communicates with the optical tracker to obtain poses of optical markers, with the remote controller of the robot arm for robot movement and feedback, and with the HoloLens for AR guidance.

Coordinate systems (COSs) involved in the proposed navigation system are summarized as follows. The three-dimensional (3D) COS of the preoperative CT image is represented by $O_{CT}$. The 3D COS of the DRB is represented by $O_{D}$. The 3D COS of the optical tracker is represented by $O_{T}$. For the robot arm, the 3D COS of the optical marker is represented by $O_{M}$. The 3D COS of the robot flange is represented by $O_{F}$. The 3D COS of the robot base is represented by $O_{B}$. For the Microsoft HoloLens 2 headset, once a HoloLens application is launched, a virtual space is defined and anchored to the environment. In this paper, we use $O_{V}$ to represent the 3D COS of the virtual space. During the surgery, the pose of the Microsoft HoloLens 2 headset relative to the virtual space COS $O_{V}$ can be tracked based on the HoloLens-SLAM algorithm [19]. Thus, a virtual model can maintain its pose in the environment when the HoloLens moves around.

Before a PAO procedure, preoperative planning is performed to generate a pelvis surface model $M^{CT}$ and to define osteotomy planes in the CT COS $O_{CT}$ (which will be introduced in Section 3.2). As shown in Figure 1, an osteotomy plane $\Psi^{CT}$ is defined by a starting point $p_{ost}^{CT}$, a normal vector $n_{ost}^{CT}$, and a horizontal vector $v_{ost}^{CT}$, which are written as outlined below:(1)$$\Psi^{CT}=\left[\begin{array}{ccc} p_{ost}^{CT} & n_{ost}^{CT} & v_{ost}^{CT}\\ 1 & 0 & 0 \end{array}\right]$$

During the PAO procedure, the proposed system aims for two tasks: one is to align the bone saw with the planned osteotomy plane, and the other is to provide AR guidance for surgeons. For the first task, we first transform $\Psi^{CT}$ from the CT COS $O_{CT}$ to the robot base COS $O_{B}$ using the following transformation chain:(2)ΨB=TFB·TMF·TTMTDT·TCTD·ΨCT In this transformation chain, TCTD is the transformation from the CT COS $O_{CT}$ to the DRB COS $O_{D}$, which is estimated using an image–patient registration (which will be introduced in Section 3.3). TDT is the transformation from the DRB COS $O_{D}$ to the optical tracker COS $O_{T}$. TTM is the transformation from the optical tracker COS $O_{T}$ to the optical marker COS $O_{M}$. Both TDT and TTM can be derived from the application programming interface (API) of the optical tracker at any time. TMF is the transformation from the optical marker COS $O_{M}$ to the robot flange COS $O_{F}$, which can be estimated using our previously published hand–eye calibration method [27]. TFB is the transformation from the robot flange COS $O_{F}$ to the robot base COS $O_{B}$, which can be derived from the API of the robot arm at any time.

Then, the robot arm is controlled to align the medical bone saw with the transformed osteotomy plane $\Psi^{CT}$. Specifically, as shown in Figure 1, the medical bone saw $\Phi^{M}$ is defined by the blade center $p_{saw}^{M}$ and a normal vector $n_{saw}^{M}$, and a horizontal vector $v_{saw}^{M}$ (which will be introduced in Section 3.4), which is written as
(3)$$\Phi^{M}=\left[\begin{array}{ccc} p_{saw}^{M} & n_{saw}^{M} & v_{saw}^{M}\\ 1 & 0 & 0 \end{array}\right]$$ Thus, the alignment between the medical bone saw and the osteotomy plane is formulated as
(4)T^FB·TMF·ΦM≡ΨB
where T^FB is the target pose of the robot arm.

For the second task, the goal is to determine the poses of the osteotomy plane, the pelvis model, and the medical bone saw in the virtual space, such that virtual models rendered by the Microsoft HoloLens 2 headset are superimposed on the corresponding physical counterparts. The poses of the osteotomy plane and the pelvis model in the virtual space COS $O^{V}$ are calculated by
(5)MVΨV=TTV·TDT·TCTD·MCTΨCT
where TTV is the transformation from the optical tracker COS $O_{T}$ to the virtual space COS $O_{V}$, which is calculated using an AR registration (which will be introduced in Section 3.5). Meanwhile, we can calculate the pose of the bone saw in the virtual space COS by
(6)ΦV=TTV·TMT·ΦM During the AR guidance, we update TDT and TMT from the API of the optical tracker at every moment. Thus, $M^{V}$, $\Psi^{V}$, and $\Phi^{V}$ are dynamically updated, allowing for the virtual models to follow the motion of both the patient anatomy and the medical bone saw.

### 3.2. Preoperative Planning

The goal of preoperative planning is to generate the pelvis model $M^{CT}$ and osteotomy planes from the preoperative CT image. We first segment the pelvis in the CT image using a threshold-based segmentation combined with a region growth algorithm [21]. Then, $M^{CT}$ is generated based on the marching cube algorithm [28]. Subsequently, osteotomy planes are manually defined on $M^{CT}$, as shown in Figure 2. For each plane, we denote its four corner points as $a_{11}^{CT}$, $a_{12}^{CT}$, $a_{21}^{CT}$, and $a_{22}^{CT}$, respectively. An osteotomy area is then defined based on the four points. We can also define a local COS, whose origin and three axes are calculated by
(7)$$o_{ost}^{CT}=\frac{a_{11}^{CT}+a_{12}^{CT}}{2}$$
(8)$$x_{ost}^{CT}=\frac{a_{21}^{CT}-a_{11}^{CT}}{∥a_{21}^{CT}-a_{11}^{CT}{∥}_{2}}$$
(9)$$z_{ost}^{CT}=\frac{x_{ost}^{CT}\times \left(a_{12}^{CT}-a_{11}^{CT}\right)}{∥x_{ost}^{CT}\times \left(a_{12}^{CT}-a_{11}^{CT}\right){∥}_{2}}$$
(10)$$y_{ost}^{CT}=z_{ost}^{CT}\times x_{ost}^{CT}$$ We can define $\Psi^{CT}$ by setting $p_{ost}^{CT}=o_{ost}^{CT}$, $n_{ost}^{CT}=z_{ost}^{CT}$, and $v_{ost}^{CT}=y_{ost}^{CT}$.

### 3.3. Image–Patient Registration

In order to estimate TCTD, an image–patient registration is performed before the osteotomy procedure, which consists of two steps. (1) Landmark-based initialization: We use the anterior inferior iliac spine, the pubic tubercle, and the ischial spine as three landmarks for initialization. Specifically, we extract coordinates of the three landmarks in the CT COS $O_{CT}$ and digitize their coordinates in the DRB COS $O_{D}$ using a trackable pointer. Then, an initial transformation can be calculated via paired-point matching [29], which roughly aligns the pelvis surface model $M^{CT}$ with the patient anatomy. (2) Iterative closest point (ICP)-based refinement: After initialization, we further digitize 40 points on the pelvis surface, acquiring their coordinates in the DRB COS $O_{D}$. Then, an ICP registration is performed between the digitized points and the roughly transformed surface model to optimize the initial transformation [30]. After the two-step image–patient registration, a fine TCTD can be estimated, accurately transforming the pelvis surface model and the preoperative planning to the DRB COS $O_{D}$.

### 3.4. Bone Saw Calibration

In bone saw calibration, we aim to calibrate the blade center $p_{saw}^{M}$, the normal vector $n_{saw}^{M}$, and the horizontal vector $v_{saw}^{M}$ of the medical bone saw $\Phi^{M}$. Specifically, as shown in Figure 3a, we digitize the four corner points of the saw blade using a trackable pointer, whose coordinates in the optical marker COS $O_{M}$ are denoted as $b_{11}^{M}$, $b_{12}^{M}$, $b_{21}^{M}$, and $b_{22}^{M}$, respectively. Then, as shown in Figure 3b, a local COS can be built based on the four points, where we define its origin and three axes by
(11)$$o_{saw}^{M}=\frac{b_{11}^{M}+b_{12}^{M}}{2}$$
(12)$$x_{saw}^{M}=\frac{b_{11}^{M}-b_{21}^{M}}{∥b_{11}^{M}-b_{21}^{M}{∥}_{2}}$$
(13)$$z_{saw}^{M}=\frac{x_{saw}^{M}\times \left(b_{12}^{M}-b_{11}^{M}\right)}{∥x_{saw}^{M}\times \left(b_{12}^{M}-b_{11}^{M}\right){∥}_{2}}$$
(14)$$y_{saw}^{M}=z_{saw}^{M}\times x_{saw}^{M}$$ Subsequently, $\Phi^{M}$ can be defined by setting $p_{saw}^{M}=o_{saw}^{M}$, $n_{saw}^{M}=z_{saw}^{M}$, and $v_{saw}^{M}=y_{saw}^{M}$.

### 3.5. AR Registration

The goal of AR registration is to estimate TTV. To this end, we propose an optical marker-based method via paired-point matching, which consists of three steps.

**Step 1**: $N_{m}$ virtual models of the optical marker are loaded in the virtual space using different poses, as shown in Figure 4a. We denote the pose of the *i*-th virtual model ($1\leq i\leq N_{m}$) in the virtual space COS $O_{V}$ as TMiV. Then, for the *j*-th infrared reflective spheres of this virtual model ($1\leq j\leq N_{s}$, where $N_{s}$ is the number of spheres of a marker), we can calculate its coordinate in the virtual space COS $O_{V}$ by
(15)pijV=TMiVpjM
where $p_{j}^{M}$ is the coordinate of the *j*-th sphere in the optical marker COS $O_{M}$. Thus, we can collect a point set in the virtual space COS $O_{V}$, which is denoted as $\Omega^{V}=\left\{p_{11}^{V},\ldots ,p_{ij}^{V},\ldots ,p_{N_{m}N_{s}}^{V}\right\}$ ($1\leq i\leq N_{m}$, $1\leq j\leq N_{s}$).

**Step 2**: We align the optical marker attached to the robot flange with each virtual model, as shown in Figure 4b. Specifically, for the *i*-th virtual model ($1\leq i\leq N_{m}$), we adjust the robot pose using the robot controller, aiming to minimize the misalignment between the optical marker and the virtual model. Compared with freehand alignment, the robot arm-based alignment has two advantages. (1) Due to the accurate movement and stability of the robot arm, no hand tremble is involved during the alignment. (2) Since the marker is held by the robot arm, surgeons can observe the misalignment from different directions. Thus, compared with freehand alignment where surgeons hold the marker and can only observe the misalignment from one direction, multi-view observation enhances depth perception, which can reduce the misalignment along the depth direction. After the virtual model is aligned with the optical marker, we derive TMT from the API of the optical tracker. Then, we can calculate the *j*-th sphere center of the *i*-th virtual model ($1\leq i\leq N_{m}$, $1\leq j\leq N_{s}$) in the optical tracker COS $O_{T}$ by
(16)pijT=TMTpjM

After aligning all virtual models and calculating all the $p_{ij}^{T}$, we collect another point set in the optical tracker COS $O_{T}$, which is denoted as $\Omega^{T}=\left\{p_{11}^{T},\ldots ,p_{ij}^{T},\ldots ,p_{N_{m}N_{s}}^{T}\right\}$ ($1\leq i\leq N_{m}$, $1\leq j\leq N_{s}$).

**Step 3**: After collecting $\Omega^{V}$ and $\Omega^{T}$, TTV is estimated by
(17)TTV=argminT1NmNs∑i=1Nm∑j=1Ns∥T·pijT−pijV∥22
where $p_{ij}^{T}\in \Omega^{T}$ and $p_{ij}^{V}\in \Omega^{V}$ ($1\leq i\leq N_{m}$, $1\leq j\leq N_{s}$). Specifically, the transformation is solved using the paired-point matching algorithm [29].

### 3.6. Robot-Assisted AR-Guided Osteotomy

After surgical planning, calibration, and registration, the proposed system can provide surgeons with both robot assistance and AR guidance during the PAO procedure.

#### 3.6.1. Robot Assistance

During the PAO procedure, the robot assistance consists of two parts: (1) **Bone saw alignment**: Given an initial pose of the robot arm, a target robot pose is calculated using Equation (Equation 4). Then, the robot arm is controlled to move toward the target robot pose, aligning the medical bone saw with the planned osteotomy plane. (2) **Osteotomy under virtual constraints**: After the alignment, the osteotomy is performed with robot assistance under virtual constraints, i.e., surgeons can freely drag the bone saw along the osteotomy plane to cut bones, but they are not able to drag the bone saw along the normal direction of the osteotomy plane or to rotate the bone saw. This is accomplished by setting the robot arm into the force mode [31]. Specifically, we first transform $x_{ost}^{CT}$, $y_{ost}^{CT}$, and $z_{ost}^{CT}$ defined in Section 3.2 from the CT COS $O^{CT}$ to the robot base COS $O^{B}$:(18)xostByostBzostB000=TFB·TMF·TTMTTD·TCTD·xostCTyostCTzostCT000 Then, using the API of the robot arm, we set the robot arm to be compliant along $x_{ost}^{B}$ and $y_{ost}^{B}$, while it is set to be non-compliant along $z_{ost}^{B}$. We also set the rotation along any axis to be non-compliant. By doing so, the motion of the bone saw can be restricted to the planned plane, providing virtual constraints for osteotomy in order to improve surgical safety.

#### 3.6.2. AR Guidance

In previously reported robot-assisted systems [13,14,15,26], despite the robot assistance, surgeons still have to frequently switch their focus between the computer screen and patient anatomy, which may lower surgical safety. In contrast, the proposed system provides surgeons with two types of AR guidance to reduce sight diversion. (1) **Visualization of virtual models**: We transform the osteotomy plane $\Psi^{CT}$ and the pelvis surface model $M^{CT}$ using Equation (Equation 5) and the bone saw $\Phi^{M}$ using Equation (Equation 6). Then, as shown in Figure 5a, virtual models can be overlaid on the corresponding physical counterparts for AR visualization, providing surgeons with visual guidance when cutting complex anatomical structures around the acetabulum. (2) **Display of pose parameters**: It is critical for surgeons to know the translation and orientation of the bone saw relative to the osteotomy plane. To this end, we also display several pose parameters of the bone saw relative to the osteotomy plane next to the patient anatomy, as shown in Figure 5a. Specifically, we first transform $x_{saw}^{M}$, $y_{saw}^{M}$, and $o_{saw}^{M}$ defined in Section 3.4 to the CT COS $O^{CT}$:(19)xsawCTysawCTosawCT001=(TCTD)−1·(TDT)−1·(TTM)−1·xsawMysawMosawM001 Then, the following parameters are defined as shown in Figure 5b: *X*, *Y*, and *Z* indicate the deviation between $o_{saw}^{CT}$ and $o_{ost}^{CT}$ along $x_{ost}^{CT}$, $y_{ost}^{CT}$, and $z_{ost}^{CT}$ directions, respectively; $\phi_{A}$ is the angle between $y_{ost}^{CT}$ and ${\hat{y}}_{saw}^{CT}$, where ${\hat{y}}_{saw}^{CT}$ is the projection of $y_{saw}^{CT}$ on the $y_{ost}^{CT}-o_{ost}^{CT}-z_{ost}^{CT}$ plane; $\phi_{B}$ is the angle between $x_{ost}^{CT}$ and ${\hat{x}}_{saw}^{CT}$, where ${\hat{x}}_{saw}^{CT}$ is the projection of $x_{saw}^{CT}$ on the $x_{ost}^{CT}-o_{ost}^{CT}-z_{ost}^{CT}$ plane. By looking at *X* and *Y*, surgeons can know the position of the saw blade center on the osteotomy plane. If the saw blade center lies in the osteotomy area, *X* and *Y* are visualized in green. Otherwise, whenever surgeons drag the bone saw out of the osteotomy area, *X* and *Y* are visualized in red, serving as a warning message. Additionally, it is expected that when the bone saw is accurately aligned with the osteotomy plane, the values of *Z*, $\phi_{A}$, and $\phi_{B}$ would be small. If *Z*, $\phi_{A}$, or $\phi_{B}$ becomes larger than the corresponding threshold (5 mm for *Z*; 3° for $\phi_{A}$ and $\phi_{B}$), the AR navigation system will alert surgeons by visualizing the parameters in red, informing surgeons that a misalignment has occurred. Therefore, by superimposing virtual models onto their physical counterparts and displaying pose parameters in the surgical area, surgeons can receive real-time visual feedback without the need to switch their view from patient anatomy to the computer screen, leading to higher accuracy and safety.

## 4. Experiments and Results

### 4.1. Tasks and Evaluation Metrics

#### 4.1.1. Evaluation of AR Registration Accuracy

To evaluate the accuracy of the proposed AR registration, we compared our method with SOTA methods [18,19,21]. Figure 6 illustrates the experimental setup. Specifically, we first defined eight validation points $\left\{q_{i}^{V}\right\}$(1≤i≤8) in the virtual space. Then, for each method, we estimated a TTV and then used a trackable pointer to digitize the validation points rendered in the HoloLens, obtaining their coordinates in the optical tracker COS $O_{T}$, which are denoted as $\left\{{\hat{q}}_{i}^{T}\right\}$ (1≤i≤8). We used the mean absolute distance error (mADE) as a metric to evaluate registration accuracy, which was calculated by
(20)mADE=18∑i=18∥qiV−TTV·q^iT∥2For each method, the experiment was repeated ten times. Thus, we calculated the average value, maximum value, and minimum value of mADE achieved by each method for comparison.

Additionally, we also conducted a qualitative evaluation using a pelvis phantom attached to a DRB. Specifically, the phantom was manufactured by 3D printing based on a pelvis surface model extracted from a CT image of a patient. The image–patient registration was performed before the experiment, transforming the surface model to the DRB COS. Then, for each method, we estimated a TTV and rendered the virtual model of the pelvis in the HoloLens. For qualitative evaluation of the registration accuracy, we compare the AR misalignment achieved by each method.

#### 4.1.2. Ablation Study

We further conducted an ablation study to investigate the effectiveness of two strategies adopted in the proposed AR registration: (1) controlling a robot arm to align the optical marker and (2) observing misalignment from different directions. In the ablation study, we followed the experimental setup introduced in Section 4.1.1, while the proposed method was compared with the following two approaches. (1) **Freehand registration** (denoted as FR): The optical marker was detached from the robot flange and then held by hand to align with virtual models; (2) **Single-view robotic registration** (denoted as SRR): In this approach, the optical marker was attached to the robotic flange and aligned with virtual models by controlling the robot arm while the misalignment was observed from only one direction. In this experiment, we also adopted mADE as a metric to evaluate the registration accuracy.

#### 4.1.3. Evaluation of Time Cost of the Proposed AR Registration

In order to investigate whether AR registration will interrupt the surgical workflow, we conducted an experiment to measure the time required by the proposed AR registration. In this experiment, we performed the AR registration five times. Each time, we measured the time required by the whole registration procedure. Referring to the intraoperative registration and calibration methods introduced in the literature [32,33,34], an average time consumption of less than 5 min is acceptable, which will not interrupt surgical workflow.

#### 4.1.4. Evaluation of Osteotomy Accuracy

We finally conducted an experiment on five sheep pelvises to evaluate the osteotomy accuracy of the proposed system. For each pelvis, the experiment was carried out based on the following steps. (1) A preoperative CT image was acquired, where pelvis segmentation and surgical planning were performed. (2) We calibrated the robot system and performed the image–patient registration as well as AR registration. (3) Osteotomy was then performed under AR guidance and robot assistance. (4) After osteotomy, a postoperative CT image was acquired. Considering the thickness of the bone saw, we used following method to extract the actual osteotomy plane from the postoperative image. As shown in Figure 7a, we first extracted two planes in the image: an upper plane and a lower plane. Then, the actual osteotomy plane was calculated as the middle plane between the upper plane and the lower plane. (5) We manually extracted bone surface models from preoperative and postoperative CT images, respectively. Then, an ICP-based surface registration was performed to align the two surface models [30], transforming the planned osteotomy section from the preoperative CT image to the postoperative CT image. Then, deviations between the planned and the actual osteotomy sections could be calculated.

We used three evaluation metrics to validate the osteotomy accuracy: center translation error (denoted as $d_{c}$), maximum distance (denoted as $d_{m}$), and angular deviation (denoted as θ), as shown in Figure 7b. Specifically, $d_{c}$ measured the Euclidean distance between the center of the planned osteotomy section and that of the actual osteotomy section. $d_{m}$ measured the maximum distance between the planned and actual osteotomy sections. θ was the angle between the normal vectors of the two osteotomy sections.

### 4.2. Experimental Results

#### 4.2.1. Accuracy of AR Registration

Experiment results of AR registration are presented in Table 1 and Figure 8. Table 1 illustrates the average value, maximum, and minimum of mADE achieved by different AR registration methods [18,19,21]. The proposed method achieved an average mADE of 1.96 ± 0.43 mm and a maximum mADE of 2.71 mm. Compared with the second-best method, our method achieved a 4.45 mm improvement in terms of average mADE. Additionally, we illustrate qualitative results in Figure 8, where virtual models (yellow) were overlaid on the pelvis phantom (white) using TTV estimated using different methods. In this figure, we used red arrows to highlight the misalignment. Compared with other methods, our method achieved the best alignment accuracy. Experimental results demonstrated the AR registration accuracy of the proposed method.

#### 4.2.2. Ablation Study

We summarize the quantitative results of the ablation study in Table 2. Compared with freehand registration (referred as FR), single-view robotic registration (referred as SRR) achieved a slight improvement, demonstrating the effectiveness of the robot arm when aligning virtual models with the optical marker. Additionally, when using multi-view observation, the average mADE was improved by a margin of 4.58 mm compared with single-view robotic registration. Such improvement demonstrated the importance of observing misalignment from different directions during the AR registration.

#### 4.2.3. Time Cost of the Proposed AR Registration

We summarized the time cost of the proposed AR registration in Table 3. As shown in the table, the proposed method achieved an average time cost of 203.6 ± 11.0 s. Thus, the time required by the AR registration is acceptable in the surgery.

#### 4.2.4. Osteotomy Accuracy

Experimental results of osteotomy accuracy are presented in Table 4 and Figure 9 and Figure 10. In this experiment, on average, the osteotomy section is 15.80 mm long and 8.56 mm wide. The proposed system achieved an average $d_{c}$ of 0.96 ± 0.23 mm, an average $d_{m}$ of 1.31 ± 0.20 mm, and an average θ of 3.77 ± 0.85°. Qualitative results are illustrated in Figure 9, where actual osteotomy planes and planned osteotomy planes are visualized in orange and yellow, respectively. Experimental results demonstrated the osteotomy accuracy achieved by the proposed system.

Additionally, in Figure 10, we show the AR guidance during the osteotomy procedure. As shown in the figure, virtual models of the sheep pelvis, the bone saw, and the planned osteotomy plane were accurately superimposed on their corresponding counterparts. At the same time, pose parameters of the bone saw were visualized in the surgical area, offering real-time feedback without sight diversion. When the bone saw was out of the osteotomy area, as shown in Figure 10a, parameters were displayed in red. Otherwise, they were visualized in green, as shown in Figure 10b.

## 5. Discussion and Conclusions

In this paper, we proposed a robot-assisted AR-guided surgical navigation system for PAO, consisting of a robot arm holding a medical bone saw, an optical marker, and a Microsoft HoloLens 2 headset. In order to provide AR guidance, an optical marker-based AR registration method was proposed to estimate a transformation from the optical tracker COS to the virtual space COS, allowing virtual models to be aligned with the corresponding physical counterparts. Additionally, for osteotomy guidance, the robotic system automatically aligned the bone saw with planned osteotomy planes and then provided surgeons with virtual constraints in order to improve surgical safety. Furthermore, in order to provide visual feedback without sight diversion, AR guidance was launched during the whole procedure, leading to higher osteotomy accuracy and safety. Comprehensive experiments were conducted to evaluate both AR registration accuracy and system accuracy. As shown in Table 1 and Table 4, the proposed AR registration method achieved an average mADE of 1.96 ± 0.43 mm, while the robotic system achieved an average $d_{c}$ of 0.96 ± 0.23 mm, an average $d_{m}$ of 1.31 ± 0.20 mm, and an average θ of 3.77 ± 0.85°. Experimental results demonstrated the AR registration accuracy and the osteotomy accuracy of the developed system.

In comparison with other SOTA methods, our AR registration method offers three advantages. (1) We take advantage of stable robot movement to align virtual models with the optical marker. Compared with freehand alignment [19,21], no hand tremble is involved. (2) During alignment, the optical marker is held by the robot arm. Thus, the proposed method allows for multi-view observation during the alignment, improving the depth perception. Compared with single-view methods [18,19], our method can achieve higher AR registration accuracy. (3) Different from the methods introduced in [19,21], our method does not require a registration tool that needs to be calibrated before the AR registration. Thus, the calibration error is not coupled with the registration in our method. These advantages are confirmed by the experimental results shown in Table 1 and Figure 8, where the proposed method achieved the best AR registration accuracy with an average mADE of 1.96 ± 0.43 mm.

The advantage of the proposed robot-assisted AR-guided surgical navigation system for PAO lies in the fact that our system combines surgical navigation, robot assistance, and AR visualization, where these three components complement each other. In particular, surgical navigation provides real-time tracking of instruments and patient anatomy, obtaining transformations between different COSs. However, it cannot provide any robot assistance. In contrast, the surgical robot provides not only accurate positioning of the bone saw but also virtual constraints during the procedure, leading to higher accuracy than freehand osteotomy. However, despite robot assistance, surgeons still have to frequently switch their focus between the surgical area and the computer screen, which may reduce surgical safety. To solve this problem, AR visualization is introduced to the system in order to provide real-time visual feedback without sight diversion. This allows surgeons to focus only on the surgical area, which may improve the safety of the PAO procedures. Another advantage of AR visualization is to serve as a sanity check of the proposed navigation system during surgical procedures. For example, an intraoperative contamination of retro-reflective spheres in optical markers can be caused by patient blood, leading to inaccurate optical tracking [35]. When AR visualization is not utilized, virtual models are visualized on the computer screen rather than over the surgical scene, making it hard for surgeons to notice any accidents. However, with the AR visualization of virtual models, surgeons can easily notice the misalignment between virtual models and the corresponding physical counterparts. Thus, surgeons can stop the osteotomy, ensuring surgical safety. Overall, the proposed system takes advantage of real-time tracking, accurate positioning in combination with virtual constraints, and visual feedback without sight diversion, holding potential for higher accuracy and safety than existing surgical robots.

It is worth discussing the limitations of our study. First, the transformation estimated by the proposed AR registration is only valid for the ongoing HoloLens application [24,36]. Similar to most outside–in AR navigation methods reported in the literature [18,19,21], the proposed AR registration is required to be performed again when a new HoloLens application is launched, which is inconvenient compared with inside–out AR navigation. Nevertheless, our method outperforms inside–out methods in the following aspects. (1) The external tracker has a larger tracking range than cameras on the OST-HMD. (2) In order to achieve high tracking accuracy, the size of fiducial markers used in inside–out methods is usually large (e.g., at least 12 cm width for a Vuforia Image Target [24]), which may intervene with surgeons’ operations due to the limited exposure of the surgical area. In contrast, this is not a problem for the proposed system, which depends on the real-time tracking offered by the optical tracker. (3) In our method, AR tracking can be decomposed into two parts: AR registration, which aims to compute TTV, and object pose tracking achieved by the optical tracker. AR registration is required to be performed only one time at the beginning of the surgery. Then, for each trackable object whose pose is obtained by the optical tracker, it can be transformed to the virtual space based on the estimated TTV, achieving AR tracking. Compared with inside–out AR tracking where marker detection, identification, and pose estimation are computed repeatedly during the whole procedure, our method can achieve not only higher tracking accuracy (especially out-of-plane accuracy) but also better robustness and faster response frequency. Additionally, as shown in Table 3, we found that the average time consumption of the proposed AR registration method was 203.6 s, which is considered acceptable according to the literature [32,33,34]. Second, although the proposed system achieved a low $d_{c}$ and $d_{m}$, such results were obtained in a laboratory environment. Additionally, the osteotomy experiment was conducted on sheep pelvises that were much smaller than the human pelvis, leading to shorter osteotomy depth. Thus, the accuracy may not reach the same level when translating this prototype system to a PAO procedure in an actual clinical scenario. However, such experimental results were still able to demonstrate the efficacy of the developed robot-assisted AR-guided surgical navigation system for PAO. Our future work will focus on evaluate the prototype system on human cadavers.

In summary, we proposed a robot-assisted AR-guided surgical navigation system for PAO. Results obtained from our comprehensive experiments demonstrated the efficacy of the proposed system.

## Figures and Tables

**Figure 1 sensors-24-04754-f001:**
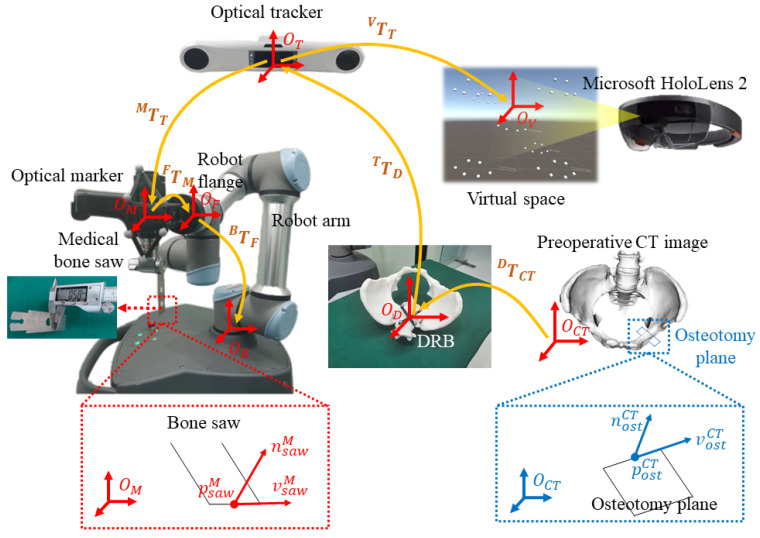
An overview of the proposed robot-assisted AR-guided surgical navigation system for PAO.

**Figure 2 sensors-24-04754-f002:**
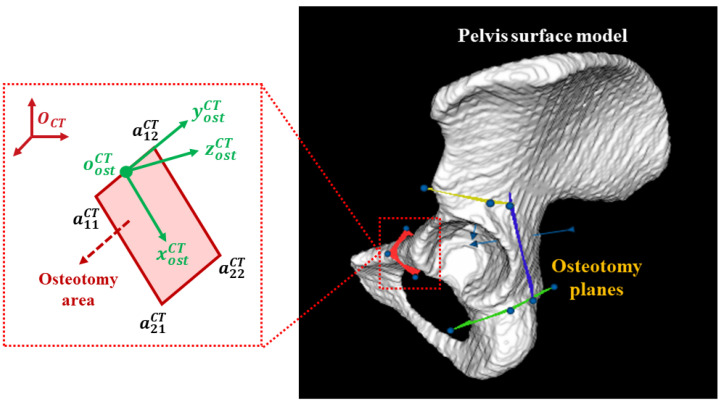
A schematic illustration of preoperative planning, where $o_{ost}^{CT}$, $x_{ost}^{CT}$, $y_{ost}^{CT}$, and $z_{ost}^{CT}$ are calculated based on $a_{11}^{CT}$, $a_{12}^{CT}$, $a_{21}^{CT}$, and $a_{22}^{CT}$.

**Figure 3 sensors-24-04754-f003:**
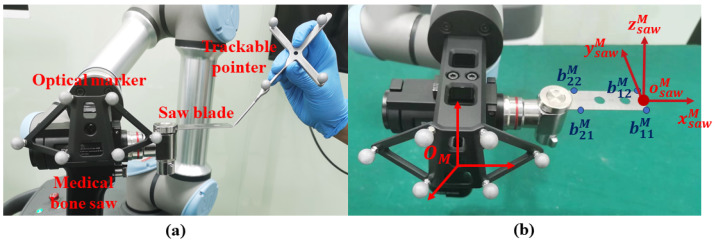
A schematic illustration of bone saw calibration. (**a**) Digitizing the four corner points using a trackable pointer; (**b**) calculating $o_{saw}^{M}$, $x_{saw}^{M}$, $y_{saw}^{M}$, and $z_{saw}^{M}$ based on $b_{11}^{M}$, $b_{12}^{M}$, $b_{21}^{M}$, and $b_{22}^{M}$.

**Figure 4 sensors-24-04754-f004:**
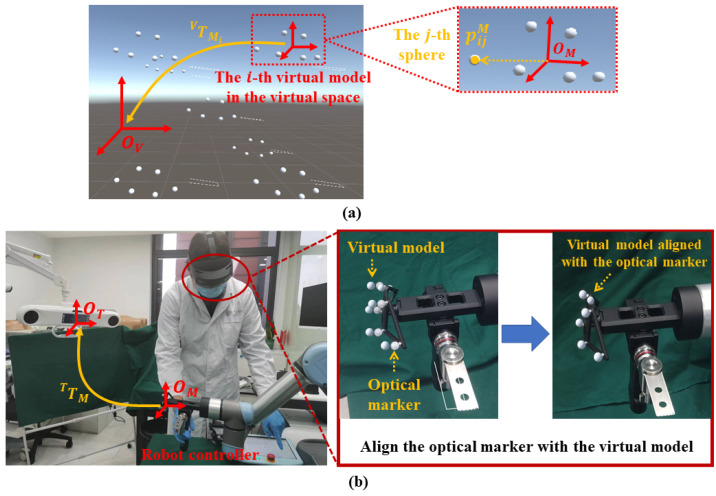
The proposed AR registration. (**a**) Virtual models of the optical marker are loaded in the virtual space. Each virtual model has a unique pose. (**b**) The optical marker attached to the robot flange is aligned with each virtual model.

**Figure 5 sensors-24-04754-f005:**
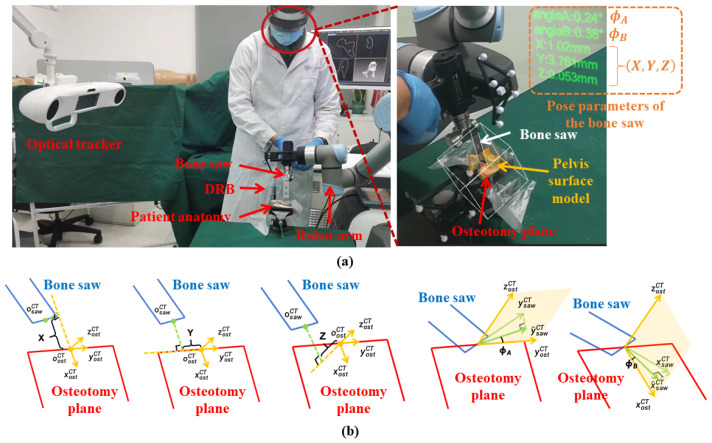
AR guidance during the PAO procedure. (**a**) The proposed AR navigation system not only provides visualization of virtual models but also dispalys the pose parameters of the bone saw relative to the osteotomy plane. (**b**) The definitions of the pose parameters.

**Figure 6 sensors-24-04754-f006:**
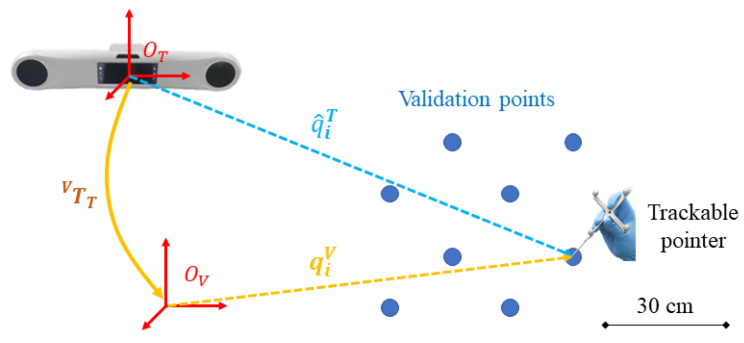
Experimental setup of the evaluation of AR registration accuracy. In this experiment, we defined eight validation points in the virtual space. Then, after performing AR registration, we used a trackable pointer to digitize the validation points, acquiring their coordinates in the optical tracker COS $O_{T}$. We calculated the mADE of the validation points as an evaluation metric.

**Figure 7 sensors-24-04754-f007:**
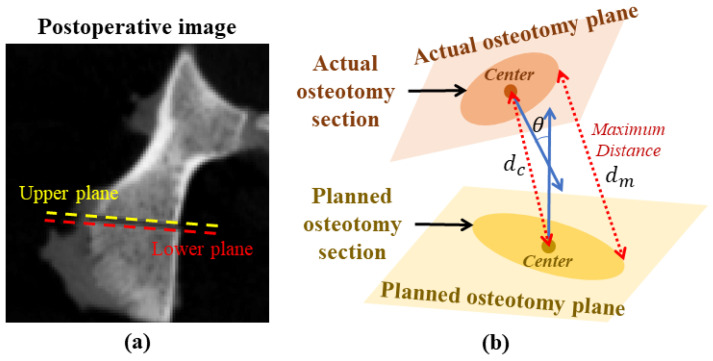
Evaluation of the osteotomy accuracy. (**a**) Extraction of the upper plane and the lower plane in the postoperative image. (**b**) A schematic illustration on how the center translation error $d_{c}$, the maximum distance $d_{m}$, and the angular deviation θ are defined.

**Figure 8 sensors-24-04754-f008:**
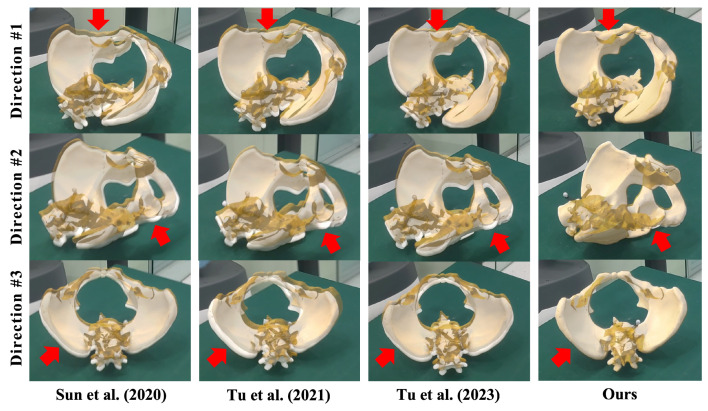
Visualization of the alignment between the virtual model (yellow) and the pelvis phantom (white) using different methods [18,19,21]. In this figure, misalignment is highlighted using red arrows. Compared with other methods, the proposed method achieved the most accurate AR registration.

**Figure 9 sensors-24-04754-f009:**
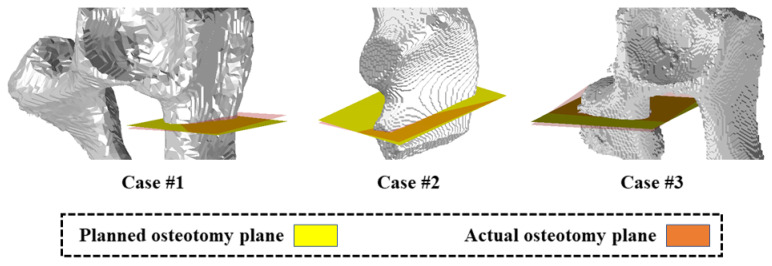
Visualization of experimental results of osteotomy where actual osteotomy planes and planned osteotomy planes are visualized in orange and yellow, respectively.

**Figure 10 sensors-24-04754-f010:**
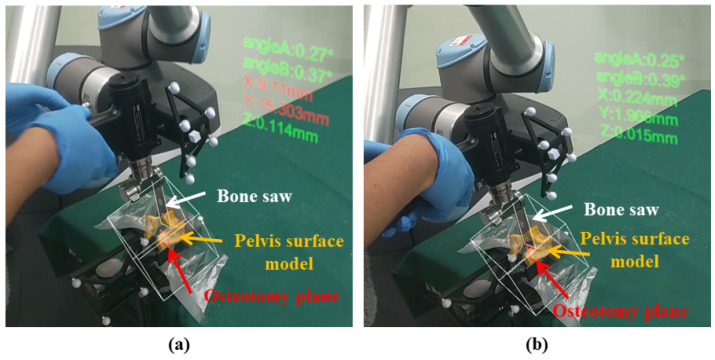
AR guidance during the osteotomy procedure: (**a**) AR display when the bone saw was out of the osteotomy area, where the pose parameters are displayed in red; (**b**) AR display when the bone saw was on the planned plane and in the osteotomy area, where the pose parameters are visualized in green.

**Table 1 sensors-24-04754-t001:** Comparison with SOTA methods for AR registration.

Method	mADE (mm)		
	**Average**	**Max**	**Min**
Sun et al. [18]	7.98±6.41	20.53	1.83
Tu et al. [19]	7.08±2.64	10.91	3.41
Tu et al. [21]	6.41±1.85	9.53	3.94
Ours	1.96±0.43	**2.71**	**1.38**

The best results are displayed in bold.

**Table 2 sensors-24-04754-t002:** Experimental results of the ablation study. (FR: freehand registration; SRR: single-view robotic registration).

	Strategies		mADE (mm)		
	Controlling Robot Arm	Multi-View Observation	Average	Max	Min
FR			6.58±2.17	9.55	3.66
SRR	*√*		6.54±3.39	11.65	2.23
Ours	*√*	*√*	1.96±0.43	**2.71**	**1.38**

The best results are displayed in bold.

**Table 3 sensors-24-04754-t003:** Time cost of the proposed AR registration.

Index	1	2	3	4	5	Average
Time(s)	201	206	190	198	223	203.6 ± 11.0

**Table 4 sensors-24-04754-t004:** Osteotomy accuracy achieved by the proposed robot-assisted AR surgical navigation system.

Case	$d_{c}$ (mm)	$d_{m}$ (mm)	θ (°)
1	1.14	1.42	4.77
2	0.92	1.27	3.14
3	1.30	1.63	4.79
4	0.71	1.08	2.75
5	0.75	1.16	3.41
Average	0.96 ± 0.23	1.31 ± 0.20	3.77 ± 0.85

## Data Availability

The original contributions presented in the study are included in the article, further inquiries can be directed to the corresponding author.

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
