# Peer review of "Robot-Assisted Augmented Reality (AR)-Guided Surgical Navigation for Periacetabular Osteotomy"

_sensors, 2024, doi:10.3390/s24144754_

Round 1

Reviewer 1 Report

Comments and Suggestions for Authors

This papers describes a prototype of a system to robotically guide bone resection for PAO with a coupled AR vision.

The paper is well structured and clear containing many technical details on the implementation and the experiments done, anyway:

-          -The system is proposed as new given that at the state of the art there are robots or navigators some of them with AR but no coupled systems integrating robots and AR. But the two systems are functionally and technically decupled (except for the virtual to real registration) without any potential advantages for the user. If the robotic guidance is so accurate, what are the advantages to use AR too? I suggest to resubmit the paper taking into account only the robot (if the authors can motivate any advancement in respect to the state of the art).

-          -The hololens to robot registration is trivial and long. It should be done every time the hololens is turned on. This registration is also user dependent given that the alignment in the OST depends also on the position of the eye. See for example [Condino, Sara, Turini, Giuseppe, Parchi, Paolo D, Viglialoro, Rosanna M, Piolanti, Nicola, Gesi, Marco, Ferrari, Mauro, Ferrari, Vincenzo (2018). How to Build a Patient-Specific Hybrid Simulator for Orthopaedic Open Surgery: Benefits and Limits of Mixed-Reality Using the Microsoft HoloLens. JOURNAL OF HEALTHCARE ENGINEERING,] and [Ferrari V., Carbone M., Condino S., Cutolo F. (2019). Are augmented reality headsets in surgery a dead end?. EXPERT REVIEW OF MEDICAL DEVICES,]. How system could be translated in the daily clinical practice given the tedious and long registration time? (3 minutes are in my opinion optimistic)

-          -The method to evaluate the cutting accuracy in terms of mean distance is wrong or not well explained. The preperative and postoperative control points should be univocally determined and coupled to be able to calculate an error on each couple (Euclidean distance) and then to offer a significative statistics. Furthermore, a mean translation error of only 0.82 mm seems a sort of miracle given the many transformations involved, the flexibility of the mechanics comprising the saw, and that a  mean angular error of 3.77° at 5 cm ( a reasonable length of the osteotomy) determines a misalignment of 3.3 mm. The estimation of osteotomy accuracy should be better explained starting from the pre to post CT registration (which impact on the final error) and offering a clear estimation of the translation error. A good approach could be to estimate the translation error in the center of the osteotomy section and the maximum distance between the sections consisting in the planned and real osteotomy.  

-          -How the CT to bone ICP registration is initialized?

-          -In the abstract “virtual-physical registration” should be changed with AR registration (more clear reading only the abstract), while “system achieved an average distance deviation of …” should be focalized/explicited   on the robotic guidance accuracy.

-         - At row 46-48 “In the literature, AR guidance in CAOS can be 47 roughly divided into two categories: ArUco marker-based methods [12,13] and external 48 tracker-based methods [14–17]” should be rewritten with a “division” in terms of tracking modalities which in general are inside-out (as the aruco marker but not only) or outside-in (the expertal tracker).

-          -The introduction does not mention the using of surgical templates eventually placed with AR to performe PAO as [Mendicino, A R, Condino, S, Carbone, M, Cutolo, F, Cattari, N, Andreani, L, Parchi, P D, Capanna, R, Ferrari, V (2022). Augmented Reality as a Tool to Guide Patient-Specific Templates Placement in Pel [...] AL INTERNATIONAL CONFERENCE OF THE IEEE ENGINEERING IN MEDICINE AND BIOLOGY SOCIETY]

-          -The error “0.27mm” at row 315 is probably a typo.

Comments on the Quality of English Language

acceptable

Reviewer 2 Report

Comments and Suggestions for Authors

This work proposes a robot-assisted augmented reality (AR)-guided surgical navigation system for Periacetabular osteotomy (PAO) surgery. For AR guidance, the authors have also proposed an optical marker-based virtual-physical registration method to estimate a transformation from the optical tracker coordinate system (COS) to the virtual space COS. Comprehensive experiments have been conducted to evaluate both virtual-physical registration accuracy and osteotomy accuracy of the developed navigation system. This paper is well-structured and well-written. Before publication, there are some questions to be solved.

1. The authors have provided some modeling of the navigation system, what is the relationship between modeling and simulation/experimental results? Is there any theoretical calculation of the established modeling?

2. The author mentioned “Compared with conventional surgical navigation, AR guidance can reduce surgeons’ sight diversion.”, more state-of-the-art continuum robot can be cited: DOI: 10.34133/cbsystems.0098; DOI: 10.34133/cbsystems.0063; DOI: 10.34133/cbsystems.0062.

3. How to measure the position and posture of the robot during the experiment? How to verify the accuracy of the characterization system?

4. The authors are suggested to provide some scale bars in the figure 6, which would help audience straightly recognized the size.

5. How this proposed method can benefit current surgical robot field. The authors are recommended to provide more discussion.

Round 2

Reviewer 1 Report

Comments and Suggestions for Authors

The authors strongly improved following the reviewers comments.

Anyway:

 A)      I still have concerns about the usefulness of the integration of the robot with AR. The authors state that a robot with a standup monitor “surgeons need to frequently switch their view between the computer screen of the navigation system and the patient anatomy” and so they integrated the Hololens AR view to:

1) display the planned cutting plane over the real scene to verify if the real saw is rightly placed. But given that the AR registration accuracy is worse than the robot one (about 2 mm Vs 1 mm) how this AR sanity check can be useful?

2) show the current robot pose in respect to the optimal one (green Vs Red) as depicted in Fig. 10. I suppose that during the automatic robot movement these numbers are red and they become green when the robot reached the planned pose. I didn’t understand: a) what is the usefulness of these information if the robot automatically place the saw without any surgeon intervention (except the movement along x and y axes)?  b) in figure 10 the errors in green are small but they should be 0 in case of a complete automatic positioning. Are this information useful in case of deviations of the saw and the robot imposed by the surgeon?  If yes, this concept should be outlined.

 B)     Given the errors measured in along the cutting plane (Dc and Dm and angle) it would be useful to know the dimensions of the osteotomy.

3     C)  Pag. 3 “coordiante”

4      D)    At page 4 an “Or” reference system is mentioned but no “Or” is in figure 1.

5     E)  How the thickness of the saw is taken into account in the evaluation of the cutting error? In general, the saw thickness is about 2 mm, bigger than the errors obtained in this study.

Reviewer 2 Report

Comments and Suggestions for Authors

The authors have addressed all comments raised by reviewers, this paper is ready for publication.
